# OpenReview forum: "TraSCE: Trajectory Steering for Concept Erasure"
_ICLR.cc/2026/Conference — Submitted to ICLR 2026_

### Official Review · Reviewer_tTZr · 2025-10-20

**Soundness:** 3
**Presentation:** 4
**Contribution:** 3
**Rating:** 6
**Confidence:** 4

**Summary:**

The authors propose a method to erase harmful content from diffusion models. They identify a corner case where vanilla negative prompting fails and replace it with a safer variant that better suppresses the targeted concept. They also introduce a localized, loss-based guidance term that nudges sampling toward the unconditional trajectory when the prompt behaves like the banned concept. The paper evaluates the approach on adversarial attack benchmarks and reports results for erasing artistic styles and objects.

**Strengths:**

1. The paper tackles a timely and important safety challenge: removing harmful content from generative models.
2. The paper is clearly written, easy to follow, and the proposed method is both simple and elegant.

**Weaknesses:**

1. The method assumes that falling back to the unconditional prior is a safe default, which may not always hold. If the base model’s unconditional distribution already leans toward risky or unwanted content in certain contexts, anchoring to it won’t prevent leakage. The approach can fail in such cases.
2. I also have sme concerns regarding the generation quality.  In diffusion, coarse scene layout forms early, while objects solidify mid-trajectory and fine details emerge late. So, when it comes to erasing the objects, the loss in the paper only activates when $$ \hat{\epsilon}_\theta(x_t, e_p) - \hat{\epsilon}_\theta(x_t, e_{np}) $$
 becomes small, typically mid/late. By that time, the scene and object placement are largely decided. Nudging the latent then can remove or weaken the target object only by distorting the image. This may be true even for complex safety concepts. For instance, the qualitative result shown in Figure 4, where the authors aim to “erase a car,” looks distorted and challenges the claim that the method preserves generation quality.

**Questions:**

1. Can the authors provide a plot that shows how the difference term varies across timesteps during the erasure of objects? This may give some clarity on the concern in 2 above.

---

> ### Author Response · Authors · 2025-11-22
> **Rebuttal to reviewer**
>
> We would like to thank the reviewer for their constructive feedback and thoughtful questions and for taking the time to review our manuscript. We have addressed their questions below.
>
> **[W-1] The method assumes that falling back to the unconditional prior is a safe default, which may not always hold. If the base model’s unconditional distribution already leans toward risky or unwanted content in certain contexts, anchoring to it won’t prevent leakage. The approach can fail in such cases.**
>
> Thank you for your insightful comment. This would only become an issue when we attempt to erase a major portion of the model's learned distribution, but in such a case, we should rather retrain the model itself, as it means the majority of the training set used is harmful content. Erasure is generally adopted for targeted minor concepts in the overall learned concept space.
>
>
> **[W-2] I also have sme concerns regarding the generation quality. In diffusion, coarse scene layout forms early, while objects solidify mid-trajectory and fine details emerge late. So, when it comes to erasing the objects, the loss in the paper only activates when $$ \hat{\epsilon}\theta(x_t, e_p) - \hat{\epsilon}\theta(x_t, e_{np}) $$ becomes small, typically mid/late. By that time, the scene and object placement are largely decided. Nudging the latent then can remove or weaken the target object only by distorting the image. This may be true even for complex safety concepts. For instance, the qualitative result shown in Figure 4, where the authors aim to “erase a car,” looks distorted and challenges the claim that the method preserves generation quality.**
>
> For concepts such as artistic styles and nudity, it would be sufficient to only apply a high erasure during the later stages of the denoising process. On the other hand, for objects, we would require higher steering in the early/middle stages. Although we have not explored adaptive losses in this paper, this can definitely be an interesting follow-up study.
>
> Furthermore, we have included a study in Appendix I to analyze the diffusion trajectory and demonstrate the benefits of the updated negative prompting strategy and TraSCE's loss-based guidance on the diffusion trajectory, even when performing object erasure.
>
> **[Q-1] Plot of the difference term across timesteps during the erasure of objects**
>
> We thank the reviewer for their thoughtful suggestion. We have incorporated a study on this in Appendix I, where we demonstrate benefits of the updated negative prompting strategy and TraSCE's loss-based guidance on the diffusion trajectory, even when performing object erasure.
>
>
> We would again like to thank the reviewer for their helpful suggestions which have helped us in improving our paper.

---

> > ### Author Response · Authors · 2025-11-28
> >
> > Thank you very much again for your insightful comments. We're happy to continue the conversation at any point throughout the discussion period. Thanks!

---

### Official Review · Reviewer_xZU9 · 2025-10-29

**Soundness:** 3
**Presentation:** 3
**Contribution:** 2
**Rating:** 4
**Confidence:** 3

**Summary:**

The paper proposes TraSCE, an inference-time method for concept erasure in text-to-image diffusion. It combines (i) a modified negative prompting formulation (replacing the conventional “negative as base” with an unconditional base plus a difference term) and (ii) a localized loss–based guidance that nudges the denoising state away from regions aligned with the undesired concept each step. Experiments report lower attack-success rates on several adversarial NSFW/violence benchmarks and show applications to artistic styles and object erasure.

**Strengths:**

1.  Addresses safety in diffusion models, especially robustness to prompt‑based jailbreaks.
2. Integrates a geometric “trajectory steering” view into the diffusion process, offering intuitive control over latent evolution.

**Weaknesses:**

1. While “trajectory steering” offers a coherent new perspective, the implementation closely resembles classifier-free or loss-based guidance mechanisms already explored in prior works (e.g., SLD). The novelty primarily lies in problem framing and loss design rather than theoretical advancement.
2. The per-step gradient update increases sampling time by 2–3×, which may limit deployment for large-scale or real-time use.

**Questions:**

1. How robust is the method to partial or semantically related prompts (e.g., “bikini” vs. “swimsuit”)? Does the trajectory steering generalize to paraphrased adversarial prompts?
2. Can the authors quantitatively characterize how the diffusion trajectories differ between baseline CFG, SLD, and TraSCE?
5. How does TraSCE behave when multiple negative concepts are specified simultaneously (e.g., “no nudity, no gore”)? Does the trajectory steering remain stable?

---

> ### Author Response · Authors · 2025-11-22
>
> We thank the reviewer for the constructive feedback and for their time and effort in reviewing our manuscript. We have addressed their questions below.
>
> **[W-1] While “trajectory steering” offers a coherent new perspective, the implementation closely resembles classifier-free or loss-based guidance mechanisms already explored in prior works (e.g., SLD). The novelty primarily lies in problem framing and loss design rather than theoretical advancement.**
>
> We would like to clarify that TraSCE first starts by identifying and fixing a problem with the traditional classifier-free guidance, which had previously gone unnoticed. Furthermore, unlike SLD and other prior works, TraSCE further enhances the modified negative prompting approaches using a loss-based guidance. SLD and other works, such as SAFREE, that work on inference either rewrite a prompt or steer the trajectory simply using the noise prediction from the negative prompt. TraSCE goes beyond this and actually analyzes the effect of noise predictions to come up with a more effective way to utilize them. This is also clearly visible in the results, where we show that TraSCE can significantly reduce the nudity production rate by over 20\% even compared to SLD-Max and SAFREE. And it does so without impacting general image quality, in contrast to SLD-Max.
>
> **[W-2] Inference cost**
>
>
> Thank you for your comment. We would like to mention that on a more recent DiT-based model, FLUX.1-dev, our approach only increases the inference time by 1.6x. The smaller increase in this case, compared to SDv1.4, would be attributed to more efficient backpropagation through the DiT blocks during the loss-based guidance. Additionally, methods that apply adaptive gradient steps or approximate gradients can be used to reduce computational costs. However, this is out of the scope of this work.
>
> **[Q-1] Robustness to partial or semantically related prompts (e.g., “bikini” vs. “swimsuit”)**
>
> We thank the reviewer for their suggestion. We have conducted experiments comparing "bikini" and "swimsuit", and included the results in Appendix H of the updated paper, where we demonstrate that TraSCE is highly robust to paraphrased prompts. We would also like to mention that it is not only robust to paraphrased prompts but also to adversarial prompts that do not directly imply the concept, as shown in Tables 1, 2, and 3 and Figures 2, 3, 4, and  6.
>
> **[Q-2] Characterization of the diffusion trajectory in comparison to CFG, SLD and TraSCE**
>
>
> We thank the reviewer for their thoughtful suggestion. We have incorporated a study on this in Appendix I, where we demonstrate benefits of the updated negative prompting strategy and TraSCE's loss-based guidance on the diffusion trajectory in comparison to CFG and SLD.
>
>  **[Q-3] Multi-concept erasure**
>
> We thank the reviewer for their suggestion. We have included an example of this in Appendix G of the updated paper, where we show that TraSCE can effectively handle multiple concepts without harming generation quality.
>
> We again thank the reviewer for their helpful suggestions which have helped us in honing our paper.

---

> > ### Author Response · Authors · 2025-11-28
> >
> > Thank you very much again for your insightful comments. We're happy to continue the conversation at any point throughout the discussion period. Thanks!

---

### Official Review · Reviewer_WyCx · 2025-10-31

**Soundness:** 2
**Presentation:** 3
**Contribution:** 2
**Rating:** 4
**Confidence:** 5

**Summary:**

This study introduces TraSCE, a method aimed at mitigating the generation of harmful content (e.g., sexual elements) in T2I diffusion models. The authors propose guiding the diffusion trajectory away from problematic generation paths. The proposed method combines two part: a modification of traditional negative prompting based on the classifier-free guidance, and a localized loss-based guidance, which further enhance the guidance performance. The performance of the proposed method is outstanding, demonstrating effectiveness on multiple models and concept erasure tasks.

**Strengths:**

1. The method is very clear and easy to understand.
2. The proposed method performs excellently and shows outstanding results on multiple evaluation benchmarks.
3. The authors' experimental setup is comprehensive, taking into account various evaluation tasks, erasure robustness, and different base models.

**Weaknesses:**

1. The application of the proposed method seems to be based on an unreasonable setting: that the specific category of harmful content must be predefined for the current generation. This is impractical in real-world scenarios. In contrast, recent related works [1, 2, 3] adopt a "detect-then-erase" mechanism, which first determines if a specific concept has been generated and only then performs concept erasure. This appears to be a more reasonable setup.
2. The additional generation time introduced by the proposed method appears to be unacceptable. For instance, in a standard classifier-free guidance model, the application of Eq. 3 directly incurs an extra 50% inference time, and the feedback optimization in Eq. 4 is even more time-consuming.
3. I appreciate that the authors adapted their experiments to the new FLUX model. However, related to the previous point, the authors must report the additional inference time overhead that the proposed method introduces when applied to FLUX. Given that FLUX is a  model with a very large number of parameters (15x than sd1.4), it is difficult to believe that the optimization mechanism from Eq. 4 can be practicably applied to it.

[1] SAFREE: Training-Free and Adaptive Guard for Safe Text-to-Image and Video Generation, iclr25

[2] Detect-and-Guide: Self-regulation of Diffusion Models for Safe Text-to-Image Generation via Guideline Token Optimization, cvpr25

[3]  Localized Concept Erasure for Text-to-Image Diffusion Models Using Training-Free Gated Low-Rank Adaptation, cvpr25

**Questions:**

Please address the weaknesses

---

> ### Author Response · Authors · 2025-11-22
> **Rebuttal to reviewer WyCx**
>
> We thank the reviewer for their time and effort and for providing us with constructive feedback on our manuscript.
>
> **[W-1] Detect-and-erase is a more reasonable setup**
>
> We thank the reviewer for sharing their opinion. We would like to start by mentioning that the TraSCE loss tries to approach zero only when the prompt is not so far away from the negative prompt, thus functioning similarly to the detect-and-erase paradigm.
>
> Further, we would like to point out that all the mentioned papers require "predefining" the erasure concept and learning the relevant token mapping, LoRA for erasure at inference. Our setting is consistent with recent work [1,2,3,4,5,6,7,8,9,10], all of which predefine a concept. Additionally, our approach works completely at inference and does not even require training a LoRA or concept mapping, i.e., it is completely training-free. Lastly, our approach outperforms SAFREE on all benchmarks that we have evaluated on.
>
> We have acknowledged the mentioned papers in our `Related Work' section. Thank you for bringing these to our attention.
>
> [1] Bui et al. "Erasing undesirable concepts in diffusion models with adversarial preservation." Neurips 2024
>
> [2] Gao et al. "Eraseanything: Enabling concept erasure in rectified flow transformers." ICML 2025
>
> [3] Thakral et al. "Fine-Grained Erasure in Text-to-Image Diffusion-based Foundation Models" CVPR 2025
>
> [4] Gao et al. "Meta-unlearning on diffusion models: Preventing relearning unlearned concepts." ICCV 2025
>
> [5] Gong et al. "Reliable and efficient concept erasure of text-to-image diffusion models." ECCV 2024
>
> [6] Heng et al. "Selective amnesia: A continual learning approach to forgetting in deep generative models" Neurips 2024
>
> [7] Huang et al. "Reliable concept erasing of text-to-image diffusion models via lightweight erasers." ECCV 2024
>
> [8] Kumari et al. "Ablating concepts in text-to-image diffusion models." ICCV 2023
>
> [9] Gandikota et al. "Erasing concepts from diffusion models." ICCV 2023
>
> [10] Gandikota et al. "Unified concept editing in diffusion models" WACV 2024
>
> **[W-2] Inference cost**
>
> Thank you for your valuable comment. Methods that apply adaptive gradient steps or approximate gradients can be used to reduce computational costs. However, this is out of the scope of this work.
>
>
> **[W-3] Inference cost on FLUX**
>
> Thank you for your suggestion. We have conducted a study to get the inference time on the FLUX.1-dev model. We observed that TraSCE only increases the inference time by 1.6x for FLUX.1-dev. FLUX with TraSCE takes 47 seconds to generate one sample on average, while it takes 29 seconds with traditional classifier-free guidance on a single H100 GPU, making it only 1.6x slower. The smaller increase in this case, compared to SDv1.4 on an A100 GPU, would be attributed to more efficient backpropagation through the DiT blocks during the loss-based guidance.
>
> We would like to again thank the reviewer for their thoughtful comments and questions which have helped us in improving our paper.

---

> > ### Author Response · Authors · 2025-11-28
> >
> > Thank you very much again for your insightful comments. We're happy to continue the conversation at any point throughout the discussion period. Thanks!

---

### Official Review · Reviewer_Nzoa · 2025-11-01

**Soundness:** 3
**Presentation:** 3
**Contribution:** 2
**Rating:** 4
**Confidence:** 3

**Summary:**

The paper proposes TraSCE, a training-free method for concept erasure in text-to-image diffusion models. TraSCE combines a modified negative prompting strategy with a localized loss-based guidance to steer diffusion trajectories away from target concepts. TraSCE is evaluated on benchmarks that include adversarial prompts and tasks such as erasing NSFW content, artistic styles, and specific objects, achieving state-of-the-art reductions in attack success rates while preserving image quality across unrelated concepts.

**Strengths:**

**S1:** TraSCE operates at inference time, eliminating the need for costly retraining or data collection. This makes it easily deployable for model owners to adapt to new concepts.

**S2:** TraSCE shows significant reductions in attack success rates against black-box adversarial attacks, with minimal degradation in general image quality.

**S3:** Experiments cover diverse erasure tasks using multiple metrics, providing a broad assessment of the method's applicability.

**Weaknesses:**

**W1:** The main concern about this paper is its limited novelty and insufficient distinction from prior work. The core component of TraSCE, the modified negative prompting, is adapted from Liu et al. (2022) on concept negation but lacks adequate justification for its novelty. While the addition of localized loss-based guidance is claimed as new, it fails to be differentiated from existing guidance techniques, such as classifier guidance. For instance, Schramowski et al. (2023) also employ trajectory steering for safety, but TraSCE's ablation studies only compare against basic negative prompting, lacking a deep analysis of how the loss function advances beyond prior art. Therefore, the authors should conduct a comparative analysis with recent inference-time methods, highlighting theoretical or empirical distinctions, and perform ablations to quantify the independent contribution of the loss guidance versus the negative prompting.

**W2:** The experimental results lack dynamic adversarial testing, white-box attack settings, and human evaluation. The paper does not test against adaptive white-box attacks that exploit TraSCE's gradient information, thereby limiting its claims of robustness. For artistic style and object erasure, assessments depend solely on automated classifiers without human evaluation or diversity metrics, risking overestimation of erasure effectiveness due to dataset biases. The authors should incorporate dynamic adversarial testing, add human evaluations for subjective tasks (e.g., artistic styles), and report diversity scores to ensure that erasure does not harm output variety.

**W3:** TraSCE increases inference time by approximately 2.6 times, which could hinder real-time applications. Hyperparameters are task-specific and tuned empirically, yet no sensitivity analysis or guidance for adaptive selection is provided. To enhance practicality, the authors could benchmark TraSCE on resource-constrained devices and suggest approximations (e.g., reducing gradient steps or using sparse updates).

**Questions:**

See Weaknesses.

---

> ### Author Response · Authors · 2025-11-22
> **Rebuttal to reviewer Noza (1/n)**
>
> We thank the reviewer for their time and effort and for giving us important suggestions on improving the manuscript.
>
> **[W-1.1]The main concern about this paper is its limited novelty and insufficient distinction from prior work. The core component of TraSCE, the modified negative prompting, is adapted from Liu et al. (2022) on concept negation but lacks adequate justification for its novelty.**
>
> We would like to clarify that the core component of TraSCE is in \textbf{first identifying the issue with conventional negative prompting, which has been used by all previous approaches without modification} such as [1,2,3]. This finding can be applied to such methods, and we believe that it can contribute to this line of research. We do not claim that Liu et al. (2022)'s formulation itself is a contribution of ours. On the other hand, we propose combining a loss-based guidance to improve upon the modified negative prompting.
>
> [1] Yoon et al. "Safree: Training-free and adaptive guard for safe text-to-image and video generation." ICLR 2025
>
> [2] Meng et al. "Concept Corrector: Erase concepts on the fly for text-to-image diffusion models"
>
> [3] Shin et al. "Prompt-Based Safety Guidance Is Ineffective for Unlearned Text-to-Image Diffusion Models"
>
> **[W-1.2] While the addition of localized loss-based guidance is claimed as new, it fails to be differentiated from existing guidance techniques, such as classifier guidance.**
>
> Classifier guidance requires a classifier in the loop to compute loss values. TraSCE requires no such classifier but rather can achieve SOTA-level results using just the noise predictions. Furthermore, we have already shown comparisons with classifier guidance in Table 8, where we compare it with using a NudeNet and CLIP model as a classifier for guidance. The results shown in the table indicate that the loss-based guidance outperforms the classifier guidance.
>
>
> **[W-1.3] For instance, Schramowski et al. (2023) also employ trajectory steering for safety.**
>
> Yes, TraSCE is based on trajectory steering, but so are almost all inference-time approaches on diffusion models. This should not imply a lack of novelty. We believe it is, in fact, even more novel that our method can utilize the same three noise predictions more effectively, as shown by the decrease in ASR from 23.16\% to 1.05\% on K77, even compared to SLD-Max.
>
> **[W-1.4] TraSCE's ablation studies only compare against basic negative prompting, lacking a deep analysis of how the loss function advances beyond prior art. Therefore, the authors should conduct a comparative analysis with recent inference-time methods, highlighting theoretical or empirical distinctions, and perform ablations to quantify the independent contribution of the loss guidance versus the negative prompting.**
>
> Our analysis tables (Tables 1, 2, 5, and 7, as well as Figures 2 and 4) include empirical comparisons with negative prompting (SD-NP), SLD, and SAFREE, demonstrating that TraSCE outperforms them. Table 6 presents an ablation study to quantify the independent contribution of each component. Furthermore, we have added an in-depth analysis of how each element affects the diffusion trajectory, as detailed in Appendix I.
>
> **[W-2] Dynamic adversarial testing and human evaluations for subjective tasks**
>
> Thank you for your suggestions. Our experimental setting is in line with recent literature, such as [1,2,3], which have all utilized static adversarial datasets to evaluate the effectiveness of their approaches. We have used the same evaluation methods, including GPT-4o for artistic styles and a pretrained classifier for object detection, in line with these papers. Furthermore, TraSCE being an inference-time method, we do not expect it to be utilized in a setting where the adversary would have white-box access to the weights.
>
> Lastly, as per your suggestion, we are conducting human evaluations on erasing artistic styles and will update the paper with these results as soon as they are available.
>
> [1] Yoon et al. "Safree: Training-free and adaptive guard for safe text-to-image and video
> generation." ICLR 2025
>
> [2] Gong et al. "Reliable and Efficient Concept Erasure of Text-to-Image Diffusion Models" ECCV 2024
>
> [3] Huang et al. "Receler: Reliable Concept Erasing of Text-to-Image Diffusion Models via Lightweight Erasers" ECCV 2024

---

> > ### Author Response · Authors · 2025-11-22
> > **Rebuttal to reviewer Noza (2/n)**
> >
> > **[W-3] Inference time**
> >
> > Thank you for your suggestions. These are interesting suggestions on how to reduce the inference cost of TraSCE. Although we have not conducted experiments or incorporated recent methods for reducing inference cost, these can definitely be adopted. This can definitely be an interesting follow-up study to optimize TraSCE to take advantage of its concept erasure gains and reduce its computational overhead.
> >
> > We would like to mention that we have conducted additional analysis on a more recent DiT-based model, FLUX.1-dev, our approach only increases the inference time by 1.6x. The smaller increase in this case, compared to SDv1.4, would be attributed to more efficient backpropagation through the DiT blocks during the loss-based guidance.
> >
> > We again thank the reviewer for their constructive feedback.

---

> > > ### Author Response · Authors · 2025-11-28
> > >
> > > Thank you very much again for your insightful comments. We're happy to continue the conversation at any point throughout the discussion period. Thanks!

---

> ### Author Response · Authors · 2025-12-02
> **Rebuttal to reviewer Noza: Additional experiments on human evaluation**
>
> **[W-2]** We conducted a human evaluation on artistic style erasure in **Appendix J**, which shows that users preferred TraSCE's outputs in terms of both erasure and image quality.
>
> | Category                                   | TraSCE | ESD | SLD |
> | ------------------------------------------ | :------------: | :------------: | :------------: |
> | **Least Resembles Van Gogh (Style)**       |   **45.56\%** |    17.93\%   |   36.41\%    |
> | **Best Image Quality**                     |   **50.54\%** |    33.15 \%   |    16.30 \%    |

---

### Author Response · Authors · 2025-11-22
**Message to the reviewers**

We thank all the reviewers for their effort and time in reviewing our manuscript and for their helpful and insightful comments. We would like to start by thanking them for appreciating that "TraSCE works at inference eliminating the need for retaining and data collection" (Nzoa), that the "proposed method performs excellently and shows outstanding results on multiple evaluation benchmarks" (WyCx, Nzoa, xZU9), and that our "experimental setup is comprehensive, taking into account various evaluation tasks, erasure robustness, and different base models" (WyCx, Nzoa).

Based on their reviews, we have made the following modifications to our paper and are currently conducting additional experiments.

1. **Appendix G (xZU9)**: We have included results on TraSCE's performance on multi-concept erasure in Appendix G showing that TraSCE is effectively able to handle multi-concept erasure in Appendix G.

2. **Appendix H (xZU9)**: We have included results on TraSCE's performance when prompted with paraphrased prompts from its erasure concept in Appendix H, showing that TraSCE is robust to such paraphrased prompts.

3. **Appendix I (Nzoa, xZU9,  tTZr)**: We have included an analysis of the diffusion trajectory and TraSCE's loss function with denoising steps in Appendix I.

4. **Section 5.5 (Nzoa, WyCx, xZU9)**: We have added an analysis on the inference time when applying TraSCE to FLUX.1-dev, where we show that it is only 1.6x more expensive while giving significant gains in concept erasure.

We are grateful for their constructive feedback, which has significantly helped us to refine and strengthen our paper.

---

### Author Response · Authors · 2025-12-02
**Summary of Review, Rebuttal, and Discussion**

Dear Area Chairs and Reviewers,

We thank all reviewers for providing valuable feedback, which significantly strengthened the paper. While the reviewers did not participate in the discussion phase, we provided detailed one-on-one responses to their comments, and we believe these clarifications have resolved most of their concerns. We have also updated our revised manuscript with new clarification and experiments.

We summarize the discussion and key points below for each reviewer.

---

## Global Responses

**Inference Time**: We have included additional experiments to show that TraSCE increased inference time on FLUX.1-dev by only 1.6x.

**Novelty**: We have clarified that our key contributions are identifying the flaw in conventional negative prompting and proposing a new loss-based guidance strategy that combined to yield SOTA-level results.

**Characterization of diffusion Trajectory with timesteps**: We have added **Appendix I**, which shows the advantages of TraSCE and its components over SLD and conventional negative prompting.

---

## Individual Responses

### Reviewer Noza

**Human Evaluation**: We conducted a human evaluation on artistic style erasure in **Appendix J**, which shows that users preferred TraSCE's outputs in terms of both erasure and image quality.

| Category                                   | TraSCE | ESD | SLD |
| ------------------------------------------ | :------------: | :------------: | :------------: |
| **Least Resembles Van Gogh (Style)**       |   **45.56\%** |    17.93\%   |   36.41\%    |
| **Best Image Quality**                     |   **50.54\%** |    33.15 \%   |    16.30 \%    |


### Reviewer WyCx


**Detect and erase is a more reasonable setup**: We have clarified that we follow the same protocol as multiple recent papers. Further, detect-and-erase methods require predefining a concept to learn LoRA or text mappings, whereas TraSCE does not, making it easier to use.


### Reviewer xZU9

**Robustness to partial or semantically related prompts (e.g., “bikini” vs. “swimsuit”)**: We have added **Appendix H** to demonstrate that TraSCE is robust to such semantically related prompts.

**Multi-concept erasure**: We have added **Appendix G** to show that TraSCE can efficiently handle erasing multiple concepts.

### Reviewer tTZr

**Unconditional prior needs to be safe**: We clarify that this would only be the case if the model is trained on mostly unsafe images; in that case, the model should be retrained on a clean dataset.

**Timestep adaptive loss**: TraSCE is effective even without this, and this could be an interesting follow-up study.

Thank you again for your time, effort, and service to the community.

Best regards,

Authors

---

### Meta-Review · Area_Chair_BAum · 2026-01-04

**Summary:**

The paper addresses an important safety problem in text-to-image diffusion models and presents extensive experimental evaluations. However, they raise several key concerns.

- Multiple reviewers consider the novelty of the paper to be limited. The core components of the proposed method, including the modified negative prompting formulation and the trajectory-level loss-based guidance, are closely related to existing classifier-free guidance and prior training-free safety steering methods. As a result, the incremental contribution beyond existing techniques is not clearly articulated.

- Although the method demonstrates improvements on some benchmarks, the evaluation lacks dynamic adversarial testing, white-box attack settings, and human evaluation.

- It remains unclear whether the proposed approach can generalize to other diffusion models, safety concepts, or attack strategies beyond those examined in the current experimental setup.

- The method introduces a substantial inference overhead. The trade-off between the observed robustness gains and the increased computational cost is not thoroughly analyzed or justified.

**Reviewer Concerns:**

After the rebuttal, the authors clarify several implementation details and add additional experimental results. However, the main concerns remain unresolved. In particular, the novelty of the proposed method relative to prior training-free guidance and negative prompting methods is still unclear. Moreover, the issue of time complexity persists and is likely to constitute a significant barrier to the method’s practical adoption and widespread use.

**Reviewer Scores:**

I expect that most reviewers would maintain their original scores. While some reviewers might slightly increase their confidence due to the additional experiments and clarifications, the core concerns regarding novelty, time complexity, and robustness would likely prevent a significant upward change in scores.

---

### Decision · Program_Chairs · 2026-01-26

Reject